# Adaptive Lifting Index (*aLI*) for Real-Time Instrumental Biomechanical Risk Assessment: Concepts, Mathematics, and First Experimental Results

**DOI:** 10.3390/s24051474

**Published:** 2024-02-24

**Authors:** Alberto Ranavolo, Arash Ajoudani, Giorgia Chini, Marta Lorenzini, Tiwana Varrecchia

**Affiliations:** 1Department of Occupational and Environmental Medicine, Epidemiology and Hygiene, INAIL, Monte Porzio Catone, 00078 Rome, Italy; a.ranavolo@inail.it (A.R.); t.varrecchia@inail.it (T.V.); 2HRI2 Laboratory, Istituto Italiano di Tecnologia, 16163 Genova, Italy; arash.ajoudani@iit.it (A.A.); marta.lorenzini@iit.it (M.L.)

**Keywords:** adaptive lifting index, biomechanical risk assessment, inertial measurement units, manual material handling

## Abstract

When performing lifting tasks at work, the Lifting Index (*LI*) is widely used to prevent work-related low-back disorders, but it presents criticalities pertaining to measurement accuracy and precision. Wearable sensor networks, such as sensorized insoles and inertial measurement units, could improve biomechanical risk assessment by enabling the computation of an adaptive *LI* (*aLI*) that changes over time in relation to the actual method of carrying out lifting. This study aims to illustrate the concepts and mathematics underlying *aLI* computation and compare *aLI* calculations in real-time using wearable sensors and force platforms with the *LI* estimated with the standard method used by ergonomists and occupational health and safety technicians. To reach this aim, 10 participants performed six lifting tasks under two risk conditions. The results show us that the *aLI* value rapidly converges towards the reference value in all tasks, suggesting a promising use of adaptive algorithms and instrumental tools for biomechanical risk assessment.

## 1. Introduction

The Revised NIOSH (National Institute for Occupational Safety and Health, USA) Lifting Equation (RNLE) is a widely used approach for rating the biomechanical risk in lifting activities to prevent the occurrence of work-related low-back disorders (WLBDs) and verify the effectiveness of ergonomic interventions [1,2,3,4,5]. The RNLE allows the estimation of the lifting index (*LI*), a valid indicator of the risk of WLBDs, by dividing the actual object load lifted (L) by the recommended weight limit (RWL) [6,7,8,9]. The goal of the RNLE is to design lifting tasks with *LI*s less than one. The ease of use of the RNLE is a factor in its appeal among field safety experts. In fact, the task variables necessary for computing the RNLE can be determined using a weight scale, a tape measure, a goniometer, and a timer [3]. Despite the extensive use of the approach, the scientific literature attributes certain criticalities to the RNLE, mainly related to the accuracy and precision of the measurement of the task variables, which may imply job risk misidentification [3,9,10,11,12,13]. Users’ capacity to measure the RNLE task variables can be influenced by the lifting task’s complexity and duration and the appropriateness of training [12,14].

These above-mentioned criticalities could now be minimized by virtue of the availability, within the new scenario represented by Industry 4.0 [15], of instrument-based tools whose hardware and software components are represented by wearable sensor networks and an estimation engine designed on appropriate algorithms, respectively [16,17]. These tools for biomechanical risk assessment could be used for both quantitative “direct instrumental evaluations” [16,18,19,20] and “rating of standard methods” [21,22,23]. 

Regarding lifting activities, we hypothesize that the use of inertial measurement units (IMUs) and sensorized insoles or shoes and a proper design of iterative algorithms could allow the measurement of the variables required to calculate the multipliers of the RNLE (distances, displacements, angles, frequencies, and forces exchanged with the environment) during lifting activities, making it possible to calculate an adaptive-LI (*aLI*) characteristized by accuracy, precision, and time variation in association with the actual mode of execution of the lifting task. 

The objectives of this study are to present the basic concept and mathematics at the base of *aLI* calculation and to use it to estimate the reference *LI* and the error that is made.

## 2. Materials and Methods

This study was carried out under the aegis of the SOPHIA (Socio-Physical Interaction Skills for Cooperative Human-Robot Systems in Agile Production, http://www.project-sophia.eu, accessed on 29 November 2023) project.

### 2.1. Concepts: Kinematic and Kinetic Data

We conducted this study by considering signals that can be recorded in the workplace using wearable sensors that do not impede the physiological motor strategy of the workers while performing the task of lifting heavy loads. Wearable sensors include the terms “miniaturized (lightweight and small size)” and “wireless connected”.

#### 2.1.1. Kinematic Data

For the unique purpose of estimating RNLE task variables, we acquired kinematic data using IMUs’ wearable sensors.

IMUs are electronic devices that incorporate accelerometers, gyroscopes, and magnetometers, which detect the acceleration, orientation, and angular velocities of body segments and joints. Since these sensors operate on the inertia principle, the term “inertial” refers to the resistance to motion (inertia) of a free mass that has been accelerated by an external force or torque [16].

In this study we used the Xsens MVN Link system (Xsens, Enschede, The Netherlands, sampling rate 60 Hz) to record participants’ whole-body kinematics. The MVN Awinda motion analysis system includes a protocol for measuring whole-body kinematics, which consists of 17 IMUs positioned over the body, one on the head, sternum, and pelvis (at the L5/S1 level), and bilaterally on each of the following body segments: scapula, upper arm, forearm, hand, thigh, shank, and foot. To achieve proper sensor placement, Xsens MVN whole-body lycra suits in various sizes (M to XXL) were used. 

#### 2.1.2. Kinetic Data

Two dynamometric platforms (P 6000, BTS, Milan, Italy), embedded in the floor (sampling rate 680 Hz), were used to acquire ground reaction forces (GRFs).

Although in our study, we used force platforms, there are many wearable devices, including insoles and sensorized shoes, which allow for GRFs’ measurement or the measurement/estimation of only the vertical component of the GRF and weight load directly in the workplace [24,25,26]. 

Kinematic and kinetic data were acquired simultaneously, synchronizing the two systems, with a trigger signal generated by a synching device (BTS Trigger Box, BTS, Milan, Italy).

### 2.2. Mathematics Procedure

In this section, we introduce the aLI, which adapts its values with each subsequent execution of the lifting task. Starting from the LI [6,7,8,9] of the RNLE method [4,5,27], we propose an instrumented revised version that changes and improves with each subsequent lifting cycle.

The workflow of the iterative algorithm is summarized in the following flow diagram (Figure 1):

At the initialization, the number of identified lifting cycles (*n*) and the error in estimating the *aLI* (*ε*) are set to equal zero.

Each lifting cycle is determined by the identification of “start” and “stop” instants, which are automatically determined in two stages.

First, on the kinematic signal relating to the vertical component of the position of both wrists, the sections in which the trend fits a bell profile are identified (see Figure 2a), corresponding to the execution of a lifting task. Where the left and right fitted bells have very close centers in time (at less than 10 samples), the presence of a lifting cycle is identified as during lifting, the movement of the hands is symmetrical.

At this point, once the lifting cycles have been identified, within each of them, the events marking the start, end of lifting, and end of lowering are identified by finding the starting and stopping events of the segment of the wrist belonging to the above-mentioned bell that best matches a template (constructed by averaging the corresponding lifting signals acquired from numerous measurements carried out in the past in our laboratory), [19,20] by minimizing the squared Euclidean distance between the segment and the search vector (see Figure 2b). We have that:The start of lifting corresponds to the first minimum of the bell.The stop of lifting corresponds to the time instant at which the identified bell reaches its maximum value.The stop of lowering corresponds to the time instant at which the bell reaches its first minimum after the global maximum.

Starting from the identification of the first lifting cycle, all necessary multipliers of the RNLE are calculated for the calculation of the relative *n*-th cycle lifting index (*LI_n_*):(1)$${LI}_{n}=\frac{L(n)}{\mathrm{LC}\cdot \mathrm{HM}(n)\cdot \mathrm{VM}(n)\cdot DM(n)\cdot AM(n)\cdot FM(n)\cdot \mathrm{CM}}$$
where L(n) is the load lifted in kilograms. In the calculation of the ${LI}_{n}$, L(n) is not entered a priori as a constant input, but it is estimated from the ground reaction forces measured by two force platforms embedded in the floor. To calculate *L*(*n*), first, the weight of the subject at rest ($L_{human})$, before starting the lifting, is calculated. This is determined as the ratio of the vertical component of the ground reaction force (${force}_{vert\_noLoad}$) divided by the gravitational acceleration ($g=9.81\;m/s^{2}$):(2)$$L_{human}={force}_{vert\_noLoad}/g[\mathrm{kgf}]$$

The force of the lifting cycle (${force}_{lifting}$) is then calculated as the average of the vertical component of the ground reaction force of the person-load system (${force}_{human\_load}$) between the start of the n-th lifting (${start}_{n}$) and the stop of the corresponding lowering (${stop\_lowering}_{n}$):(3)$${force}_{lifting}=\frac{\sum_{i={start}_{n}}^{{stop\_lowering}_{n}}{force}_{human\_load}\left(i\right)}{{stop\_lowering}_{n}-{start}_{n}}[N]$$

The weight of the person-load system is then calculated as the ratio between the force of lifting (${force}_{lifting}$) and the acceleration of gravity:(4)$$L_{human\_load}={force}_{lifting}/g[\mathrm{kgf}]$$

The lifted weight is estimated as the difference between the weight of the Person-Loaded System and the reference weight (Table 1):(5)$$L(n)=L_{human\_load}-L_{ref}[\mathrm{kgf}]$$

*LC* is the load constant, equal to 23 kg, as required by the RNLE [5]. 

HM(n) is the horizontal multiplier calculated at the beginning of the *n*-th lifting, using the following equation [5]:(6)$$\mathrm{HM}(n)=\frac{0.25}{H({start}_{n})}=\frac{0.25}{{Hands}_{mp}({start}_{n})-{Ankle}_{mp}({start}_{n})}$$
where $H({start}_{n})\;$ is the horizontal location (Figure 3), measured in centimeters, at the start of the *n*-th lifting cycle and computed as the distance in the anterior–posterior direction between the midpoint of the left and right third knuckle (${Hands}_{mp}(n)$) and the mid-point between the ankles (${Ankle}_{mp}(n)$). To compute this parameter, as well as for the multipliers that need kinematic data about the hands, we corrected the used XSens kinematic model (see the section on kinematic and kinetic models) to obtain the wrist position, with a trigonometric approach to obtain the position of the center of the hands properly by adding the distance between the wrist and the third metacarpal from the wrist position (measured for each considered subject).

$VM\left(n\right)$ is the vertical multiplier, computed using the following equation [5]:(7)$$VM\left(n\right)=1-0.3\;\cdot \left|0.75-V\left({start}_{n}\right)\right|$$
where $V\left(n\right)$ is the vertical location (Figure 3), measured in centimeters, of the *n*-th lifting cycle, defined as the height of the hands (midpoint of the left and right third knuckle) from the floor at the beginning of n-lifting.

DM(n) is the distance multiplier, calculated using the following equation [5]:(8)$$DM(n)=0.82+\frac{0.045}{D\left(n\right)}=0.82+\frac{0.045}{\left|V({stop}_{n})-V({start}_{n})\right|}$$
where D(n) is the vertical travel distance (Figure 3), measured in centimeters, defined as the absolute value of the difference between the vertical height of the hands (midpoint of the left and right third knuckle) from the floor at the end of the *n*-th lifting task and at the beginning.

AM(n) is the asymmetry multiplier, calculated using the following equation [5]: (9)$$AM(n)=1-0.0032\cdot A(n)=1-0.0032\cdot \left|A\left({stop}_{n}\right)-A({start}_{n})\right|$$
is the difference in the angle between the subject’s sagittal plane and the center of the load at the end and beginning of the *n*-th lifting task. In the present case, all the liftings took place completely in the sagittal plane; therefore, the asymmetry angle was always equal to 0, and AM(n) was equal to 1 for each n.

CM(n) is the coupling multiplier of the *n*-th lifting task, indicating the quality of gripping and ranging between 0.9 and 1 [5]. In the present case, all the liftings took place with excellent subject–load coupling; therefore, CM(n) was always set equal to 1.

FM(n) is the frequency multiplier [5], calculated using the following equation:(10)$$FM\left(n\right)=f\left(F\left(n\right)\right)$$
where $F\left(n\right)$ is the frequency that the algorithm can automatically determine, according to the following equation:(11)$$F\left(n\right)=\left\{\begin{array}{c} 1\;if\;n=1\;\\ 60*\frac{n}{∆}\;if\;n>1 \end{array}\right.[\mathrm{lifts}/\min]$$
where ∆ is the difference between the start of the n-th lifting task (${start}_{n}$) and the start of the first lifting task (${start}_{1}$): $∆={start}_{n}-{start}_{1}$.

Once ${LI}_{n}$ has been calculated, for each lifting cycle, the adaptive-Lifting Index aLI is calculated (Figure 1) as the average of the lifting indices calculated from n=1 to the current *n*-th cycle:(12)$$aLI\left(n\right)=\frac{\sum_{k=1}^{n}{LI}_{k}}{n}$$

At the same time, the error between the current aLI and that of the previous cycle is calculated according to the formula:(13)$$\varepsilon_{n}=aLI\left(n\right)-\frac{\sum_{k=1}^{n-1}{LI}_{k}}{n-1}=aLI\left(n\right)-aLI(n-1)$$

As a condition for the convergence of the algorithm, to have an increasingly accurate estimate of the risk level, the triple constraint, with data-driven thresholds, imposed that:(I)The absolute value of the difference between the estimated frequency and the reference frequency (*F_ref_*, Table 1) is lower than ${5*10}^{-1}$ (frequency convergence);(II)The absolute value of the error at $\varepsilon_{n}$ is less than ${3*10}^{-1}$;(III)The absolute value of the ${∆}_{\varepsilon }$, defined as the difference between the error at the *n*-th cycle and that of the previous cycle (*n* − 1), is less than $10^{-2}$.

In formulas:(14)$$\begin{array}{c} \;\left|F\left(n\right)-F_{ref}\right|<0.5\;\\ AND\\ \left|\varepsilon_{n}\right|<0.3\\ AND\\ {∆}_{\varepsilon }\left(n\right)=\left|\varepsilon_{n}-\varepsilon_{n-1}\right|<0.01\;[\mathrm{lifts}/\min] \end{array}$$

As the first constraint, we selected frequency because the *LI* estimate depends a lot on the frequency with which the liftings are carried out, and it is necessary to wait a certain number of cycles to best estimate the frequency of lifting. We also worked on the error so that it could be small (II constraint) and, above all, stable (III constraint).

If this triple condition is verified, the algorithm is considered convergent, and we obtain the cycle (cycle_conv_), *aLI* (aLI_conv_), error (ε_conv_), and ${∆}_{\varepsilon }$ (${∆}_{\varepsilon -conv}$) at convergence. Otherwise, it continues identifying lifting cycles and adjusts the *aLI* iteratively as further cycles are completed until convergence is achieved. 

With each successive iteration of the calculation, new values of the task variables, multipliers, and aLI are obtained. An approximate value is determined at each lifting cycle, and these values form a succession that, if convergent, yields increasingly precise values of the aLI. 

### 2.3. Experimental Procedures

#### 2.3.1. Participants

The study recruited 10 participants (5F and 5M), with a mean age of 37.20 ± 5.15 years, a height of 1.68 ± 0.08 m, a weight of 68.35 ± 13.68 kg, and a body mass index [BMI] of 23.90 ± 3.44 kg/m^2^. All participants were not taking part in any clinical medication trials and had no history of back pain, upper or lower limb or trunk surgery, neurological or orthopaedic diseases, or problems with the vestibular system. Participants provided written informed consent for the study, which followed the Helsinki declaration and was approved by the local ethics committee (Comitato Etico “LAZIO 2”, N.0078009/2021), after receiving a thorough description of the experimental process. There was no mention of the expected results to prevent potential bias.

#### 2.3.2. Experimental Procedure

Before measurements were started, the Xsens system was calibrated for each participant, using the “N-pose and walk” technique prior to starting recordings. To record Xsens, the Xsens MVN Analyse program (version 2018.0. 0) was utilized. Furthermore, participants underwent a training session to become familiar with the assessment procedures and correctly execute lifting tasks. The participants were asked to perform a manual material lifting task standing in a neutral body position and lifting a plastic crate with handles using both hands, according to the RNLE [5]. Table 1 shows, for each lifting condition, the reference values of the load weight (*L*), the horizontal (*H*, Figure 3) and vertical (*V*, Figure 3) locations, the vertical travel distance (*D*, Figure 3), the asymmetry angle (*A*), the lifting frequency (*F*), and the corresponding reference values of the multipliers. The hand-to-object coupling was defined “good” for all lifting tasks.

The lifting tasks were designed to obtain reference *LI* values of 0.5 (tasks 1, 2, and 3 in Table 1) and 1.5 (tasks 4, 5, and 6 in Table 1). Note that tasks within the same LI (1, 2, and 3 for *LI* = 0.5 and 4, 5, and 6 for *LI* = 1.5) differed only for variables *F* and *L* (Table 1). Each participant was required to perform all six tasks, each lasting 4 min, at the frequencies described in Table 1 for each task. A metronome was utilized to trigger the lifting frequency: each time the acoustic signal was received, the participants raised the load to the set height. A rest period of 5 min was used between tasks. The lifting tasks were assigned to each participant in a random order.

#### 2.3.3. Errors of Measured Variables

To estimate the measurement errors in the variables measured with IMU and force platforms with respect to reference values (Table 1), the absolute error (E_a_) and the relative percentage error (E_r_) are evaluated at each cycle (E_aH_ and E_rH_ for H, E_aV_ and E_rV_ for V, E_aD_ and E_rD_ for D, E_aL_ and E_rL_ for L), as follows:(15)$$E_{a}=\left|{variable}_{measured}-{variable}_{ref}\right|$$
(16)$$E_{r}=100*\frac{\left|{variable}_{measured}-{variable}_{ref}\right|}{{variable}_{ref}}[\%]$$
where ${variable}_{ref}$ are the values defined in Table 1.

### 2.4. Statistical Analysis

The statistical analyses were carried out using SPSS 20.0 (IBM SPSS). The Shapiro–Wilk test was used to verify the normal distribution of the data. One sample *t*-test was used to compare the values of aLI_conv_ with the reference values for each task. Furthermore, for each LI we performed, we conducted a one-way repeated-measures ANOVA to determine whether there were any significant differences among the three tasks with the same LI (tasks 1, 2, and 3 with LI = 0.5 and tasks 4, 5, and 6 with LI = 1.5). The statistical significance level was set at α = 0.05.

## 3. Results

Figure 4 shows the box-whisker plots of the results relating to the estimated variables of the RNLE. Figure 4 shows the values of the *H*, *V*, *D*, and *L*, calculated using wearable sensors, force platforms, and the algorithm described in Section 2.2, and the reference values of the variables for each task (Table 1). Figure 5a, b shows the E_a_ and E_r_ of these variables, respectively. The absolute errors are in meters for *H*, *V*, and *D* and in kgf for *L*.

For each task, Figure 6 shows the mean and SD values of estimated F at each cycle and the mean and SD values of cycle_conv_. 

For each task, Figure 7 shows the mean and SD values of *aLI* (7a), error (7b), and Δ*_ε_* (7c) at each cycle and the mean and SD values of cycle_conv_. 

Figure 8 shows the values of aLI_conv_, cycle_conv_, ε_conv_, and Δ*_ε_*. One sample *t*-test showed no statistical significant differences between aLI_conv_ and reference values for task 1 (t = 1.0124; *p* = 0.341), task 2 (t = 1.455; *p* = 0.196), task 3 (t = 0.4035; *p* = 0.696), and task 5 (t = −1.2247; *p* = 0.2603), while statistical significant differences * were found for task 4 (t = −4.4915 p = 0.003) and task 6 (t = −4.8603; *p* = 0.001). The repeated measures ANOVA revealed no significant effect of the task on *aLI* for *LI* = 0.5 (*F* = 1.278, *p* = 0.321) and *LI* = 1. 5 (*F* = 2.815, *p* = 0.107).

More in detail, in Table 2, the mean (±SD) of aLI_conv_ and ε_conv_ for each task are reported. 

## 4. Discussion

In this study, we used the *aLI* to estimate the level of biomechanical risk in heavy lifting activities. This index receives information from a network of IMU sensors for estimating *H*, *V*, *D*, and *F* and from force platforms to estimate the weight of the lifted load. The calculation algorithm allows for an adjustment of the estimation cycle after cycle until the definitive value is identified, which occurs when three conditions are respected.

The results show us that the *aLI* value rapidly converges towards the reference value (Table 1) for all six tasks performed in the two risk conditions (Figure 7 and Figure 8). In the lowest risk condition, the tool takes from 6 to 7 cycles to 12 to 13 cycles, about 2 min, to correctly estimate *LI*, while in the highest risk condition, the cycles used fluctuate between 7 and 20, about 3 min (Figure 7). Furthermore, in the lowest risk condition, there is, on average, a slight overestimation of the risk level, which is greater in the tasks with a higher frequency (maximum overestimation for task 2 at frequency 6 liftings/min, then 1 with 4 liftings/min and then 3 with 3 liftings/min), but in all cases, it reduces as the considered number of cycles increases. Meanwhile, in the highest risk condition, in the two lower frequency tasks, 4 and 6, there is a slight underestimation of the risk level, while in task 5, initially, there is an overestimation of the risk level, which, however, is then reduced significantly as the considered cycles increase. We also observe that in tasks with a higher level of risk, there is greater intra-subject variability in the estimate of the *aLI*, which, however, at the same level of risk, is greater in tasks with a higher frequency (Figure 7). Furthermore, statistical analysis suggests that the estimation algorithm should be optimized, especially for higher risk levels (Figure 8). On the other hand, the estimate is independent of the method of execution of the lifting activity within the same risk level (Figure 8), and this allows us to be confident in the generalizability of the procedures.

These data suggest a promising use of this approach in terms of measurement times, which, excluding the worker’s dressing time, are very short. Furthermore, such a rapid estimate of the risk level suggests the use of *aLI* within control algorithms for early intervention collaborative robots to support lifting activities.

The results of this study also show that the accuracy of the estimation ranges between 0.05 and 0.18 in the lowest risk lifts and between 0.10 and 0.33 in the highest risk lifts (see Table 2). Instead, the estimation precision varies between 0.06 and 0.10 in tasks with *LI* = 0.5 and between 0.07 and 0.39 in tasks with *LI* = 1.5 (Table 2). These results are also particularly useful in understanding that this instrumental approach can minimize the misidentification of tasks [12,28] with a good classification of the same.

From a detailed analysis of the estimation made on the RNLE variables, regarding *H*, the results show an overestimation for the lifting performed in the risk condition *LI* = 0.5, with a median error that never exceeds 5 cm (Figure 4), and an underestimation for those with *LI* = 1.5, with a median error that never exceeds 10 cm. This error is fundamentally attributable to a drift in the rotation of the local reference system with respect to the laboratory one, which can be typical of IMU systems and can be minimized through several procedures [29,30,31,32]. Furthermore, it will be necessary to better estimate, compared to what was done with the correction factor, the position between the two hands when picking up the load (see the methods section). It will be particularly important to improve the estimation of *H*, as previous sensitivity analyses have shown that the horizontal position, in addition to being important for determining the *LI*, is normally, together with frequency, the variable most affected by error with traditional measurements [12]. For the estimation of the variable *V*, there is an overestimation for both the risk conditions, with errors always below 10 cm. Also, this type of error can be minimized by improving the correction procedure on the calculation of the hands’ position. The estimation of *D* is particularly accurate for both risk classes, with a slightly higher error for the one with *LI* = 1.5. In this case, any systematic errors in the measurement are eliminated in the difference. The estimate of the weight of the lifted object is accurate enough in both risk conditions, with an underestimation error almost always below 1 kgf. The type of sensor employed to detect the ground reaction forces is the most likely factor influencing the inaccuracy in the estimation of the weight of the lifted load, implying a systematic error.

Future developments include the use of countless wearable devices, insoles, and sensorized shoes, instead of force platforms, to enable GRF measurement and weight load estimates directly in the workplace [24,25,26]. However, the data on the frequency estimation are very interesting and are accurate and precise after just a few lifting cycles and therefore after about one or two minutes (Figure 6). This ability is a notable advantage compared to what is accomplished with traditional measurements, as scientific literature shows that the frequency variable is normally affected by error [12].

## 5. Conclusions

The concepts, mathematics, and results of this study suggest a very promising use of instrumental tools and automatic algorithms for rating the variables provided by the RNLE and therefore for estimating the level of risk.

Indeed, the approaches proposed in this study are very useful in new Industry 4.0 scenarios where manual material handling activities can be performed by hybrid human–robot teams [15]. In this case, the robot control algorithms can refer to these available indices in real-time and make the robots intervene when the exposure to risk is high. Furthermore, an index conceived in this way would also allow rapid administration of acoustic, visual, or vibrotactile feedback stimuli to workers to warn the workers that they are exposing themselves to a high risk [33,34]. For these reasons, there are also many experiences and activities [17] underway to include these new approaches within the International Standard Organization (ISO) 11,228 parts 1 and the technical report 12,295 [27,35].

However, the results of this study cannot be generalized as the sample tested is still too small, and the type of lifting is very far from representing the wide variability present in the workplace. The precision and accuracy of the *aLI* will have to be increased using additional correction factors for the estimation of the variables *H* and *L*. Furthermore, the ability of the *aLI* to correctly estimate the biomechanical risk level will have to be tested in further simple tasks but with different characteristics. In particular, it will be necessary to investigate asymmetric lifting in order to test a better ability of the algorithm to estimate the asymmetry variable compared to that of health and safety operators in the workplace [12,28,36,37,38]. Finally, it is important to test and refine the algorithm’s performance in challenging tasks that require the estimation of composite *LI*s (*CLI*s), variable *LI*s (*VLI*s), and sequential *LI*s (*SLI*s) [5,35].

## Figures and Tables

**Figure 1 sensors-24-01474-f001:**
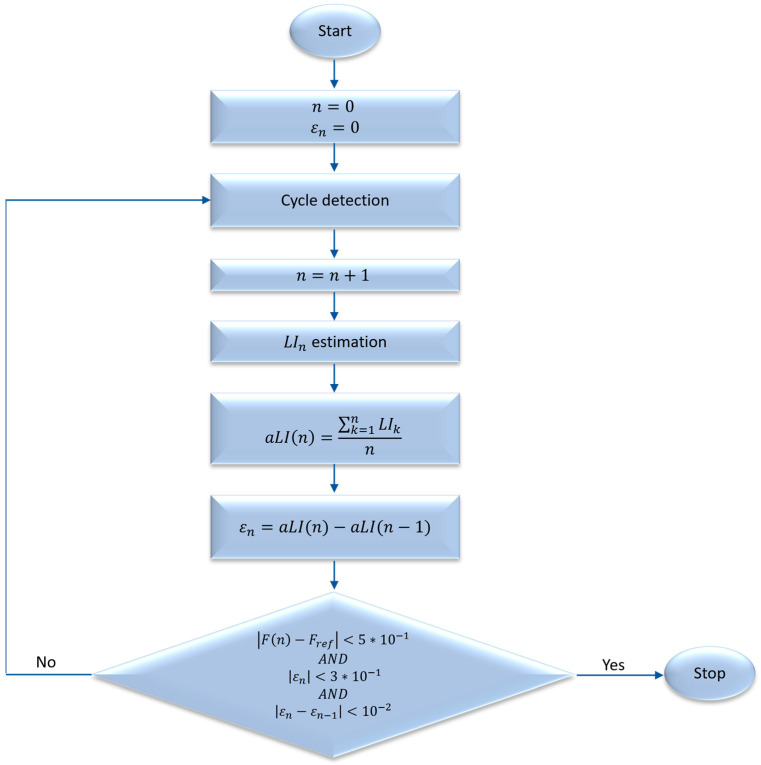
Flow diagram of the iterative algorithm to compute the adaptive lifting index (*aLI*); *n*: identified lifting cycles; *ε_n_*: error in estimating the *aLI* at *n*-th cycle; *LI_n_*: lifting index at *n*-th cycle; *F*: lifting frequency measured; *F_ref_*: reference lifting frequency.

**Figure 2 sensors-24-01474-f002:**
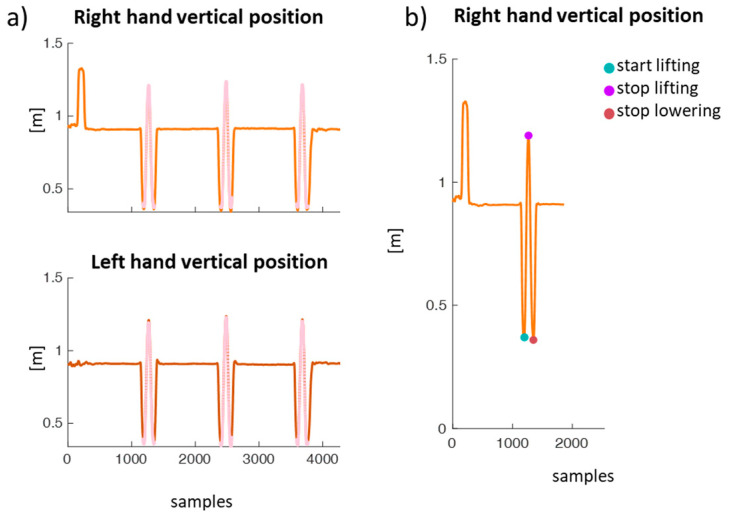
(**a**) Lifting cycles’ identification on the vertical component of the hand position for a representative participant. The sections in which the curve fits a bell profile are highlighted in pink. (**b**) Start lifting, stop lifting, and stop lowering identification on the right hand of the representative participant.

**Figure 3 sensors-24-01474-f003:**
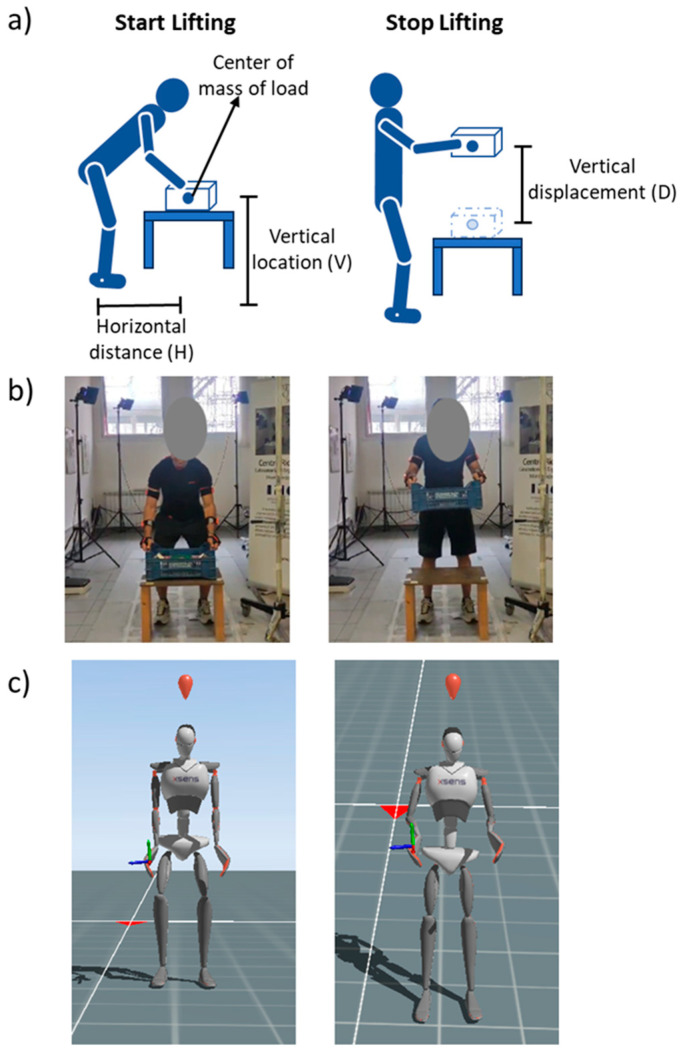
Experimental procedure: (**a**) horizontal distance, vertical location, and displacement definitions; (**b**) representative participant at start (**left**) and stop (**right**) of lifting; (**c**) kinematic reconstruction at start (**left**) and stop (**right**) of lifting.

**Figure 4 sensors-24-01474-f004:**
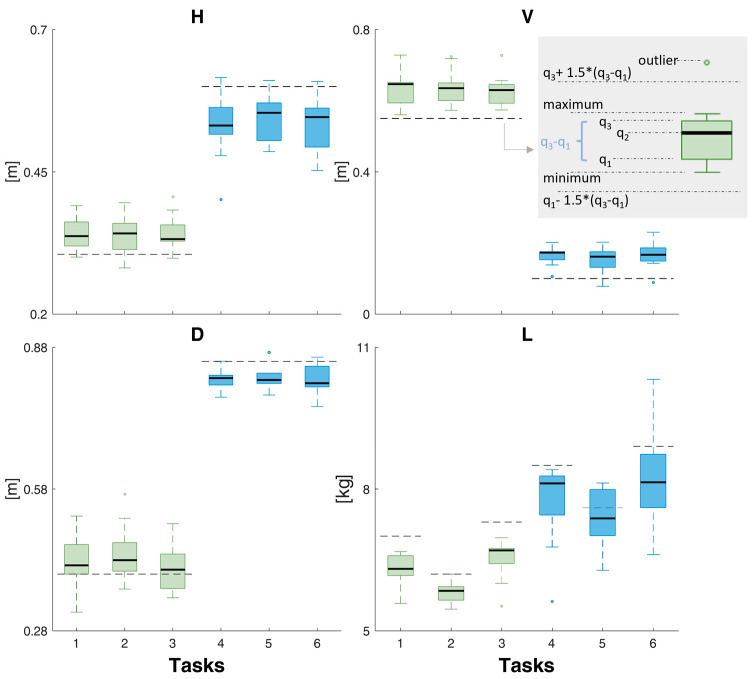
The *x*-axis of all graphs shows tasks 1 to 6. The graphs show the estimated (box-whisker plots) and reference (black dotted lines) values of the horizontal (H) and vertical (V) locations, vertical travel distance (D), and load weight (L). For each plot, the horizonal black line in the middle of each box is the sample median (q_2_), the bottom and top of each box are the 25th (q_1_) and 75th (q_3_) percentiles of the sample, respectively. The whiskers are lines extending above and below each box: whiskers go from the end of the interquartile range to the furthest observation within the whisker lengths: q_1_ − 1.5 × (q_3_ − q_1_) and q_3_ + 1.5 × (q_3_ − q_1_). The dots are the outliers (values that are more than 1.5 times the interquartile range away from the bottom or top of the box). n_sample_: number of samples (10 in this case). The gray panel provides an explanation of the plot.

**Figure 5 sensors-24-01474-f005:**
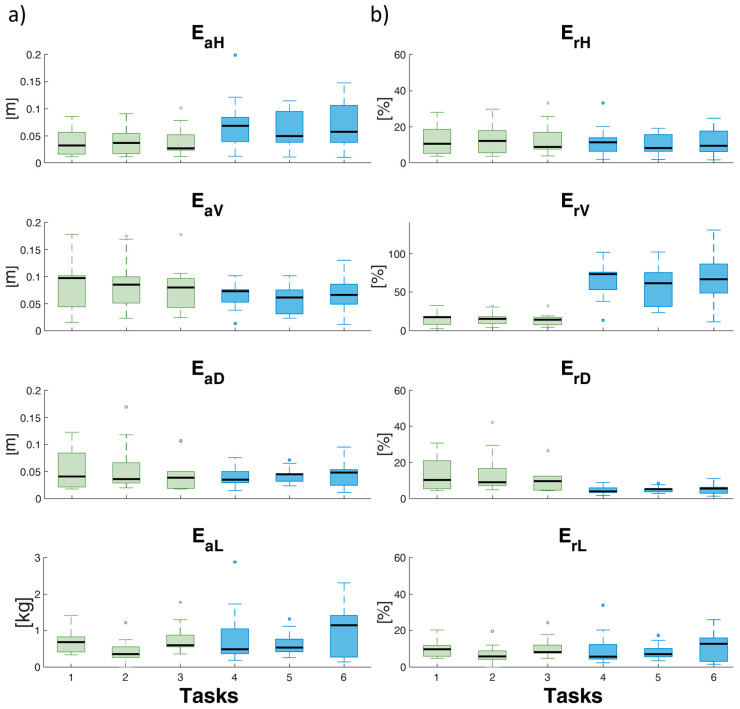
The *x*-axis of all graphs shows tasks 1 to 6. The graphs of rows (**a**,**b**) show, respectively, the absolute and relative percentage errors of the horizontal (H) and vertical (V) locations, vertical travel distance (D), and load weight (L). For each plot, the horizonal black line in the middle of each box is the sample median (q_2_), the bottom and top of each box are the 25th (q_1_) and 75th (q_3_) percentiles of the sample, respectively. The whiskers are lines extending above and below each box: whiskers go from the end of the interquartile range to the furthest observation within the whisker lengths: q_1_ − 1.5 × (q_3_ − q_1_) and q_3_ + 1.5 × (q_3_ − q_1_). The dots are the outliers (values that are more than 1.5 times the interquartile range away from the bottom or top of the box).

**Figure 6 sensors-24-01474-f006:**
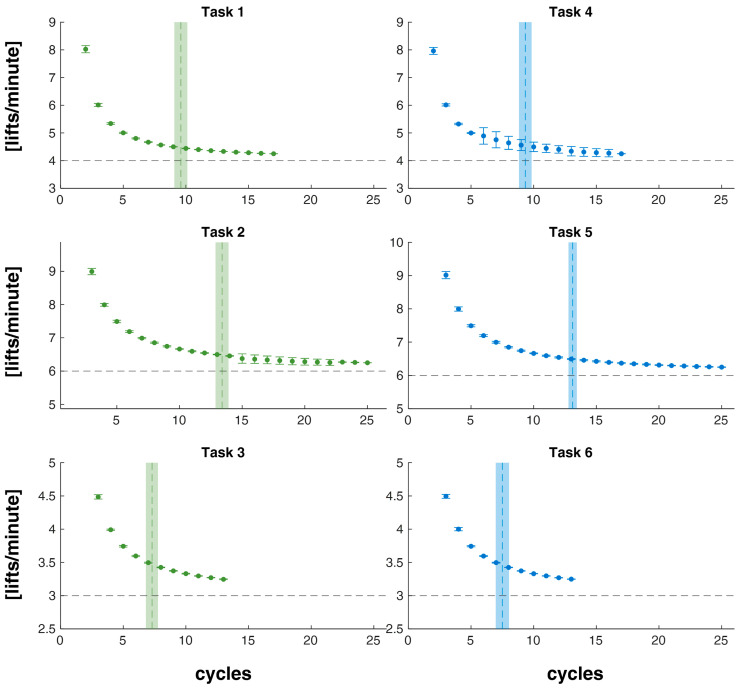
For each task (1, 2, 3, 4, 5, and 6), the mean and SD values of the estimated frequency (F) and the reference values are depicted with black dotted lines. Vertical lines are the mean and SD values of cycle_conv_ (cycle at algorithm convergence).

**Figure 7 sensors-24-01474-f007:**
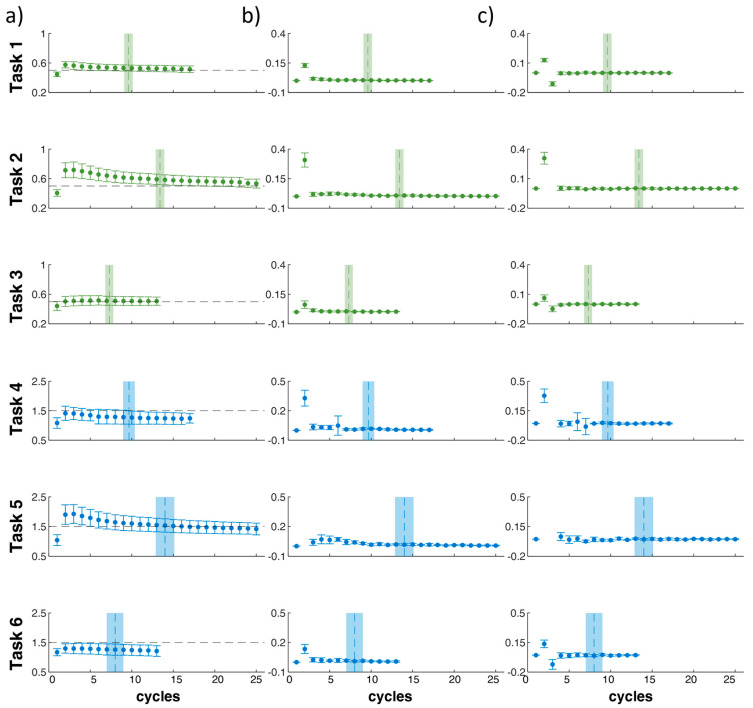
For each task (1, 2, 3 4, 5, and 6), the mean and SD values of the *aLI* (**a**), error (**b**), Δ*_ε_* (**c**) at each cycle are presented. Black dotted lines (**a**) represent the reference values of LI, as defined in Table 1. Vertical lines are the mean and SD values of cycle_conv_.

**Figure 8 sensors-24-01474-f008:**
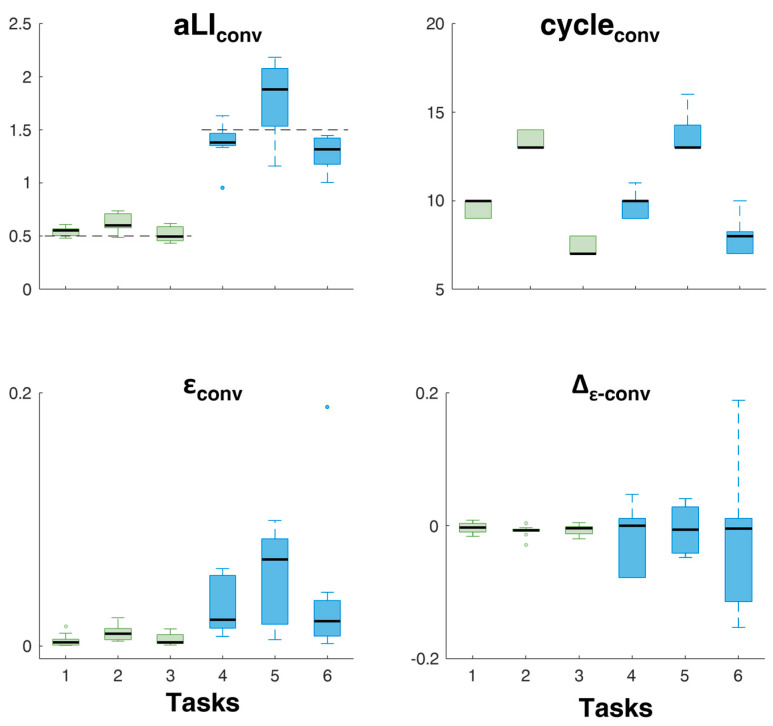
For each task (1, 2, 3, 4, 5, and 6), the aLIs at stop of algorithm (aLI_conv_), cycles at stop (cycle_con_), errors (ε_conv_) at stop of algorithm, and difference between the error at stop of algorithm and that at the previous cycle (Δ_ε-conv_). Black dotted lines represent the reference values of *LI*, as defined in Table 1. In each plot, the horizonal black line in the middle of each box is the sample median (q_2_); the bottom and top of each box are the 25th (q_1_) and 75th (q_3_) percentiles of the sample, respectively. The whiskers are lines extending above and below each box: whiskers go from the end of the interquartile range to the furthest observation within the whisker lengths: q_1_ − 1.5 ∗ (q_3_ − q_1_) and q_3_ + 1.5 × (q_3_ − q_1_). The dots are outliers (values that are more than 1.5 times the interquartile range away from the bottom or top of the box).

**Table 1 sensors-24-01474-t001:** For each task (1, 2, 3, 4, 5, and 6): the reference values of the load weight (*L*), the horizontal (*H*) and vertical (*V*) locations, the vertical travel distance (*D*), the asymmetry angle (*A*), the lifting frequency (*F*), the hand-to-object coupling (*C*), the corresponding reference values of the multipliers obtained from the Revised NIOSH Lifting Equation (see text for further details), recommended weight limit (*RWL*), and Lifting index (*LI*).

Task	*LC* (kgf)	*H* (cm)	*HM*	*V* (cm)	*VM*	*D* (cm)	*DM*	*A* (°)	*AM*	*F* (Lifts/min)	*FM*	*C*	*CM*	*L* (kgf)	R*WL*	*LI*
1	23	30.5	0.82	55	0.94	40	0.93	0	1	4	0.84	good	1	7	13.88	0.50
2	23	30.5	0.82	55	0.94	40	0.93	0	1	6	0.75	good	1	6.2	12.39	0.50
3	23	30.5	0.82	55	0.94	40	0.93	0	1	3	0.88	good	1	7.3	14.54	0.50
4	23	60	0.42	10	0.81	85	0.87	0	1	4	0.84	good	1	8.5	5.66	1.50
5	23	60	0.42	10	0.81	85	0.87	0	1	6	0.75	good	1	7.6	5.05	1.50
6	23	60	0.42	10	0.81	85	0.87	0	1	3	0.88	good	1	8.9	5.93	1.50

**Table 2 sensors-24-01474-t002:** The mean and standard deviation values (M ± SD) of aLI_conv_ and ε_conv_ for each task.

	Task					
	1	2	3	4	5	6
aLI_conv_	0.56 ± 0.06	0.68± 0.10	0.51 ± 0.06	1.43 ± 0.12	1.47 ± 0.39	1.39 ± 0.07
ε_conv_	0.01 ± 0.01	0.01 ± 0.01	0.01 ± 0.01	0.03 ± 0.02	0.14 ± 0.26	0.04 ± 0.06

## Data Availability

Data are available in a publicly accessible repository that does not issue DOIs. Publicly available datasets were analyzed in this study. These data can be found here: https://humandatacorpus.org/lifting-and-carrying-iso-11228/, accessed on 29 November 2023 [27].

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
