# Peer review of "Adaptive Lifting Index (aLI) for Real-Time Instrumental Biomechanical Risk Assessment: Concepts, Mathematics, and First Experimental Results"

_sensors, 2024, doi:10.3390/s24051474_

Round 1
Reviewer 1 Report
Comments and Suggestions for Authors
The work presented describes an interesting algorithm to calculate an adaptive lifting index (aLI) to be used in classifying lifting tasks as addressed by the RNLE, thus estimating the level of biomechanical risk in these kind of activities. The purpose is to iteratively adapt LI value in each cycle, during the motion. Data to compute aLI is obtained from external IMUs incorporated in an Xsens system, and force platforms data. Authors also defined a experimental protocol which allows to calculate parameters in no-load and load-conditions.
The protocol is very well described and and calculations clearly specified.
The algorithm is tested in a group of subjects performing 6 different tasks at 3 different frequencies. Please identify the tasks always in the same way (either 1 to 6 or A to F).
The discussion presented is very accurate and the obtained results are promising, as stated. Authors give a brief insight on the possible applications for this instrumental risk assessment tool which is very interesting and very useful on real-time monitoring workplace processes.
Conclusions section could be improved by moving 3 last paragraphs from the discussion section to this one.
Overall, the manuscript is clearly written and conclusions are supported by the results. The list of references is diversified and up to date.
Author Response
Dear reviewer,
you can find in attachment our answers.
Best regards,

Reviewer 2 Report
Comments and Suggestions for Authors
Dear Authors,
below you can find my comments:
1. In the Abstract I would add at least a sentence reporting the main results of this study.
2. Figure 2 can be improved. In particular, the panel c can show only one cycle with each relevant index marked and clearly identified by a legend or similar.
3. In the several equations present in the manuscript I would suggest also to add the measurement units.
4. About the triple constraint used to allow convergence of the algorithm, I think it would be useful to say something more on how these values were selected, otherwise it seems a bit arbitrary.
5. I think I did not well understand whether the computation of the aLI is made for every lifting cycle. Could clear this point?
6. Make uniform the definition of the 6 “tasks” or “trials”. Additionally, it is not clear what do you mean with “3 configurations for each LI”.
7. I would move the description of what is in the Figure 3 (about lines 271-280) inside the figure caption. Same comment for the other figures.
8. The algorithm takes some time, namely lifting cycles, to reach convergence. So, it works, if I understood well, only in case a worker is performing the same type of lifting, but what will happen if the worker performs mixed types of lifting? Can you discuss this aspect?
Comments on the Quality of English Language
The English can be improved
Author Response

(The authors gave the same response as above.)

Reviewer 3 Report
Comments and Suggestions for Authors
1. The introduction of figures on line 104 of page 3 and on line 119 of page 4 shall be "Figure 2b)" and "Figure 2c)" respectively.
2. On line 231 of page 7, the unit expression "kg/m2" needs properly revision;
3. Please rephrase the "Discussion" and "Conclusions" sections, ensuring a balanced transfer of more conclusive details from the former to the later.
4. For reference [17], please complete it by adding the source and year of publication. Regarding references [23, 34], include the respective page numbers of the papers.
5. Please clarify how errors are calculated in equations (15) and (16). Ensure that the error formulas are appropriate for the context, and explain why these specific formulas were chosen.
6. If applicable, consider adding information about the statistical significance of the results that demonstrated in the section 3. Indicate whether the observed differences or trends are statistically significant.
Comments on the Quality of English Language1. The introduction of fiures on line 104 of page 3 and on line 119 of page 4 shall be "Figure 2b)" and "Figure 2c)" respectively;
2. On line 231 of page 7, the unit expression "kg/m2" needs properly revision;
3. Please rephrase the "Discussion" and "Conclusions" sections, ensuring a balanced transfer of more conclusive details from the former to the later.
4. For reference [17], please complete it by adding the source and year of publication. Regarding references [23, 34], include the respective page numbers of the papers.
Author Response

(The authors gave the same response as above.)
